# The Regulatory Effect of Phytochemicals on Chronic Diseases by Targeting Nrf2-ARE Signaling Pathway

**DOI:** 10.3390/antiox12020236

**Published:** 2023-01-20

**Authors:** Wen-Jiang He, Cheng-Hao Lv, Zhong Chen, Meng Shi, Chao-Xi Zeng, De-Xing Hou, Si Qin

**Affiliations:** 1College of Food Science and Technology, Hunan Agricultural University, Changsha 410128, China; 2Infinitus (China) Company Ltd., Guangzhou 510665, China; 3College of Biological Science and Technology, Hunan Agricultural University, Changsha 410128, China; 4The United Graduate School of Agricultural Sciences, Faculty of Agriculture, Kagoshima University, Kagoshima 890-0065, Japan

**Keywords:** phytochemicals, chronic disease, Nrf2-ARE pathway, chemoprevention

## Abstract

Redox balance is essential to maintain the body’s normal metabolism. Once disrupted, it may lead to various chronic diseases, such as diabetes, neurodegenerative diseases, cardiovascular diseases, inflammatory diseases, cancer, aging, etc. Oxidative stress can cause or aggravate a series of pathological processes. Inhibition of oxidative stress and related pathological processes can help to ameliorate these chronic diseases, which have been found to be associated with Nrf2 activation. Nrf2 activation can not only regulate the expression of a series of antioxidant genes that reduce oxidative stress and its damage, but also directly regulate genes related to the above-mentioned pathological processes to counter the corresponding changes. Therefore, targeting Nrf2 has great potential for the prevention or treatment of chronic diseases, and many natural phytochemicals have been reported as Nrf2 activators although the defined mechanisms remain to be elucidated. This review article focuses on the possible mechanism of Nrf2 activation by natural phytochemicals in the prevention or treatment of chronic diseases and the regulation of oxidative stress. Moreover, the current clinical trials of phytochemical-originated drug discovery by targeting the Nrf2-ARE pathway were also summarized; the outcomes or the relationship between phytochemicals and chronic diseases prevention are finally analyzed to propose the future research strategies and prospective.

## 1. Introduction

Reactive oxygen species (ROS) and reactive nitrogen species (RNS) modulated by exogenous factors, such as ultraviolet lights, ionizing radiation, chemotherapeutics, inflammatory cytokines and environmental toxins, or by endogenous factors including mitochondria, peroxisomes, lipoxygenases, NADPH oxidase, cytochrome P450, SOD and glutathione [1,2,3], have been proposed as the main second messengers in the activation of several signaling pathways leading to mitogenesis or apoptosis [4]. While ROS or RNS is constantly generated for essential biologic functions, excess generation or an imbalance between oxidants and antioxidants can produce a common pathophysiological condition in the form of perturbations in redox circuity, known as oxidative stress [5]. Accumulated data have revealed that excess oxidative stress is closely related to many kinds of chronic diseases, such as cardiovascular diseases, cancer, neurodegenerative diseases, diabetes, obesity, ageing and other chronic inflammatory diseases [4,6,7,8].

Organisms are continuously threatened and intermittently exposed to oxidative damage caused by environmental factors. To counterbalance this, organisms and cells have created a variety of adaptation mechanisms to maintain their genomic stability. One of the most versatile mechanisms of adaptation is the Nrf2-ARE antioxidant pathway [9]. Oxidative stress may trigger the transactivation of a battery of cytoprotective genes such as antioxidant proteins, drug-metabolizing enzymes, drug-efflux pumps, heat shock proteins, 26S proteasomes, growth factors, growth factor receptors and various transcription factors, which may play important roles in DNA repair, genomic surveillance and cell growth [10]. Studies have proved that Nrf2 regulates the transcription of various antioxidant enzymes in cells by binding to ARE, as shown in our previous review [11], ARE is the promoter region of the most antioxidant and detoxifying enzymes, and Keap1 acts as a molecular switch to turn on and off Nrf2-mediated oxidation reactions. Under normal homeostasis conditions, Keap1 is in the off position and acts as an E3 ubiquitin ligase that continuously targets Nrf2 with Cul3-Rbx1 for ubiquitination and proteasomal degradation. Under oxidative stress, oxidizing agents or electrophiles modify the cysteine residue of Keap1 to release Nrf2 in the Keap1-Cul3-Rbx1 E3 ubiquitin ligase complex. Another approach is that oxidizing agents or electrophiles can phosphorylate Nrf2 through certain protein kinases, also leading to the release of Nrf2 bound to keap1. Nrf2 heterodimerizes with small Maf or other indeterminate nuclear proteins in the nucleus, binds to ARE, and eventually activates its downstream genes. 

Recent studies both in vitro and in vivo including cell models, animal models, and even human models imply that dietary phytochemicals could provide an inexpensive, adequately safe, readily applicable and easily accessible approach to activate the Nrf2-ARE pathway and exert their roles on chronic diseases control and management. Strengthening of cellular defense mechanism or restoration of stress-response signaling by administrating dietary phytochemicals provides an important strategy for chronic disease chemoprevention [12]. This article reviews the focus on the possible mechanism of Nrf2 activation by natural phytochemicals in the prevention or treatment of chronic diseases and the regulation of oxidative stress.

## 2. Phytochemicals Target Nrf2 for Diabetes Intervention

Diabetes is one of metabolic diseases in which a person has high blood sugar, either because the body does not produce enough insulin, or because cells do not respond to the insulin. This high blood sugar produces the classical symptoms of polyuria (frequent urination), polydipsia (increased thirst) and polyphagia (increased hunger) [13]. Diabetic patients have an increased risk to develop many complications such as diabetic retinopathy with progression of the disease leading to blindness and end-stage renal failure [14], cardiovascular disease (CVD) with leading atherosclerosis [15], diabetic nephropathy with leading scarring changes in the kidney tissure, loss of small or progressively larger amounts of protein in the urine, and eventually chronic kidney disease requiring dialysis [16]. During hyperglycaemia, reactive oxygen species and mitochondrial dysfunction caused excess oxidative stress that plays a causal role in the development and progression of the above diabetic complications [17]. In brief, as shown in Figure 1, ROS provoked by hyperglycemia and free fatty acids leads to activation of several signaling pathways, including NF-κB, p38 MAPK, and JNK, which cause chronic inflammation and production of a series of cytokines through suppressing secretion of insulin and promoting cell dysfunction [18].

A series of studies have showed the priority of phytochemicals in diabetic disease prevention. For example, traditional Chinese medicines (TCM) are usually served as adjuvants to improve diabetic syndromes in combination with routine antidiabetic drugs [19]. A soya phytochemical extract, namely as an oestrogenic agent, acts as an inhibitor of intestinal glucose-uptake and a preventive agent for glucose-induced lipid peroxidation [20]. As shown in Figure 1, the leaf extract of *Annona squamosa* L. (Annonaceae) showed antidiabetic activity in a rat model [21]. *Malmea depressa* administration can improve glycemic control through blocking hepatic glucose production. A medicinal plant from Thailand also showed hypoglycemic activity in normoglycemic and alloxan-induced diabetic mice [22]. However, the underlying mechanism of antidiabetic phytochemicals remained unclear.

A recent investigation using experimental diabetic Nrf2-knockout mice based on administration of streptozotocin (STZ) clearly demonstrated a protective role of the Nrf2 pathway in diabetic nephropathy [23]. This study documented that the glomeruli of patients with diabetic nephropathy were under oxidative stress and had elevated Nrf2 levels, which suggested that dietary or pharmacological activation of Nrf2 could be used as a strategy to prevent or retard the progression of this debilitating complication of diabetes in humans. Knockdown of Keap1 gene and enhancement of Nrf2 overexpression in non-obese diabetic mice can reduce the incidence of type I diabetes, which proved that Nrf2 activation effectively inhibits T cell infiltration in islets and partially restores insulin secretion function [24].

Phytochemicals show their promising potency on diabetes prevention through the activation of Nrf2-ARE pathway, which is critical for this action. Cinnamic aldehyde significantly attenuated common metabolic disorder symptoms associated with diabetes in Nrf2 (+/+) but not in Nrf2 (-/-) mice [25]. Here, we summarized the anti-diabetic effect of phytochemicals and classified them into three main approaches, which included regulation of redox balance, insulin resistance and ferroptosis- or apoptosis-induced cell death, as shown in Figure 1.

Firstly, phytochemicals can regulate redox balance by targeting Nrf2 pathway in diabetic prevention. For example, resveratrol was demonstrated to have the ability to protect diabetic kidney by attenuating markers of hyperglycemia-mediated oxidative stress and renal inflammatory cytokines via Nrf2-Keap1 signaling [26]. Quercetin and myricetin could reduce the risk of diabetic cataract formation via affecting multiple pathways, including the Nrf2 pathway [27]. Luteolin regulates NRF2-mediated oxidative stress and NF-κB-mediated inflammatory responses to protect cells from high glucose damage [28]. Geneposide inhibited ROS accumulation, NF-κB activation and inflammatory cytokine secretion in vitro and in vivo [29]. Thus, it protects the retinas of diabetic mice from developing lesions. Puerarin increased the mRNA expression levels of Nrf2 and HO-1, decreased the expression level of IL-1β, and protected the retina of STZ-induced diabetic rats [29]. Garlic sartan isopropyl ester inhibits the NF-κB signaling pathway through the activation of SIRT1/Nrf2, reduces the expressions of NF-κB p65 and inflammatory cytokines TNF-α and IL-1β, and attenuates diabetes-induced oxidative stress and inflammation in high-fat and STZ-induced rats [30].

Secondly, phytochemicals can improve insulin resistance and lower blood sugar to prevent type II diabetes. Ellagic acid, for example, inhibits Keap1 by upregulating miR-223 targeting, thereby activating Nrf2 signaling pathway, restoring the phosphorylation of IRS1/AKT and alleviating insulin resistance induced by high glucose [31]. In high-fat diet mice fed total sesquiterpenoid glycosides in Loquat leaves, Nrf2 expression was significantly increased, while serum glucose and insulin levels were decreased, and IRS-1/GLUT4 was repaired, suggesting that insulin resistance was removed and Nrf2 was involved [32]. Trilobatin significantly reduced the high fasting glucose level and improved insulin resistance of KK-Ay mice, promoted Nrf2 nucleus translocation and increased the antioxidant protein expressions of HO-1 and NQO-1, while upregulating the phosphorylation levels of Akt and GSK-3 and GLUT-2 protein expression [33].

Thirdly, recent studies found that phytochemicals can regulate ferroptosis and apoptosis to prevent diabetic complications. Nuclear translocation of NRF2 inhibits glucose production in liver cells and reduces the expression of key enzymes in gluconeogenesis [34]. Through AMPK-mediated NRF2 activation, the levels of ferritin, SLC7A11 and GPx4 are upregulated to prevent ferroptosis. If ferroptosis and apoptosis are inhibited, the body damage caused by diabetes may be prevented. For example, sulforaphane can suppress ROS-induced hyper glycemia and metabolic dysfunction in human microvascular endothelial cells and diabetic rat by activating Nrf2 pathway [35]. Recently, sulforaphane were reported to protect cardiomyocytes from the attack of diabetes, reduce the accumulation of lipid peroxide, and alleviate insulin resistance via the essential AMPK/Nrf2/GPx4 axis [36]. Besides, irisin stimulates extracellular regulatory protein kinase ERK1/2 and Nrf2 nuclear translocation phosphorylation, increases mRNA expression of HO-1, SOD1 and SOD2, and prevents apoptosis induced by high glucose and high lipid levels [37]. Honokiol activated SIRT1 signal transduction, enhanced Nrf2 nuclear translocation, and improved myocardial oxidative damage and apoptosis [38]. Naringin inhibits STZ-induced apoptosis of islet MIN6 cells by activating Nrf2 and its target genes GST and NQO1 [39]. Finally, tropaeolum majus and broccoli sprout extract have completed clinical studies about diabetes, while Chickpea and Black bean are recruiting for clinical trials, as shown in Table 1.

## 3. Phytochemicals Target Nrf2 for Neurodegenerative Diseases Intervention

Neurodegenerative diseases are causes of the progressive loss of structure or function of neurons, including death of neurons due to the expression of certain gene alleles, toxicant administration and ageing [40]. Many neurodegenerative diseases, including Parkinson’s Disease (PD), Alzheimer’s Disease (AD), Huntington’s Disease (HD) and Amyotrophic Lateral Sclerosis (ALS), occur as a result of neurodegenerative processes. The commonalities among neurodegenerative diseases include protein aggregation, proteasomal or autophagic dysfunction, inflammation, neuronal apoptosis, oxidative stress, mitochondrial dysfunction and interactions between neurons and glia [41]. Among these pathologies, the causal nature of mitochondrial dysfunction and oxidative stress in neurodegeneration is widely considered, and accumulated evidence suggests that free radicals are extremely important in causing neuronal death [42].

Consecutive and long stimulation from excess oxidative stress caused by ROS or RNS could induce damage in neurons. The central nervous system (CNS) is particularly sensitive to oxidative stress, owing to a high oxygen consumption and exposure under excess polyunsaturated fatty acids, making it particularly vulnerable to lipid peroxidation. Oxidative damage to key intracellular targets such as DNA or proteins by free radicals has been shown to be a major cause of the neuronal cell damage related to degenerative diseases [43]. In brief, as shown in Figure 2, excess oxidative stress ROS and RNS lead to aggregation and accumulation of bad proteins like α-synuclein protein of Lewy bodies, amyloid precursor protein (APP) and amyloid β peptides, which are major neuropathological alterations in neurodegenerative diseases. These proteins released from neurons lead to the activation of transcription factor NF-κB and AP-1 in microgials and astrocytes, and sequentially induce ROS, iNOS, COX-2, NAPDH oxidase, proinflammatory cytokines and inflammatory mediators, which in turn damage neurons and finally cause neurodegenerative diseases [44].

Targeting the prevention of oxidative stress and mitochondrial dysfunction shows priority on chemoprevention of neurodegenerative diseases that can be achieved through stimulation of endogenous cellular mechanisms. Nrf2-ARE antioxidant pathway is considered to play central role in prevention of neurodegenerative diseases due to their induction of gene expression of downstream phase II detoxifying enzymes and antioxidant proteins. In addition, Nrf2 inhibits the expression of BACE1, which control product of amyloid beta peptide (Aβ) as rate-limiting enzymes, thereby improving cognitive deficits in animal models of AD. A number of studies in animal models revealed that the activation of Nrf2-ARE pathway is beneficial for prevention of various diseases of the central nervous system [45]. In human AD brain, the amount of Nrf2 is reduced in the hippocampus, and histochemical analyses confirmed that Nrf2-mediated transcription is not induced in AD patients [46]. Further study on plenty of AD patients has found that common variants of the Nrf2 gene may affect disease progression, potentially altering clinically recognized disease onset [47].

Phytochemicals have been reported to activate Nrf2-ARE pathway to induce the expressions of genes that encode antioxidant enzymes, protein chaperones, phase II enzymes, neurotrophic factors, and other cytoprotective proteins, which then eliminate excess oxidative stress and reduce toxic proteins production to prevention against neurodegenerative diseases [48]. Data from epidemiological studies of human populations suggest that phytochemicals in fruits and vegetables can protect the nervous system against disease. People who consume higher than average amount of vegetables and fruits had a lower risk for Alzheimer’s disease [49]. 

Numerous studies of cell culture and animal models have demonstrated that dietary supplementation with specific fruits or vegetables, or their extracts or specific chemical components, could prevent neurodegenerative diseases. For example, as shown in Figure 2, dietary supplementation with blueberries protected dopaminergic neurons against dysfunction and degeneration in a rat model of Parkinson’s disease, and improved learning and memory without affecting amyloid pathology in a mouse model of Alzheimer’s disease [50]. Drinking pomegranate juice reduced the amount of amyloid and improved behavioral deficits in a mouse model of Alzheimer’s disease [51]. Moderate consumption of red wine reduced amyloid pathology in a mouse model of Alzheimer’s disease [52]. For specific phytochemicals, toxic protein production can be reduced and activation of the Nrf2-Keap1-HO-1 signaling pathway is observed. Sulforaphane showed its neuroprotective effect in animal and cell models of neurodegenerative condition [53], and it can activate Nrf2 to inhibit BACE1 mRNA expression and reduce Aβ production in the prevention of Alzheimer’s disease [54]. Anthocyanins were detected to activate the Nrf2/HO-1 pathway, attenuating elevated ROS levels and oxidative stress in APP/PS1 mice and hippocampal HT22 cells induced by β-amyloid oligomerism (AβO) [55]. Oxyphylla A reduced the expression levels of amyloid precursor protein (APP) and amyloid beta protein (Aβ) alleviated cognitive decline in SAMP8 mice [56]. Curcumin, resveratrol, and green tea catechins revealed a positive relationship between consumption of these compounds and the prevention of AD [57], accompanying a reduction of the formation of neurotoxic β-amyloid fibrils [58].

Amazingly, phytochemicals were also found to exert their chemopreventive effects on neurodegenerative diseases by inhibiting ferroptosis via activation of the Nrf2 pathway. As shown in Figure 2, gastrodin protects nerve cells by up-regulating the Nrf2/HO-1 signaling pathway to inhibit iron death [59]. Forsythoside A inhibits iron deposition and lipid peroxidation by activating the Nrf2/GPX4 axis and reduces iron death. Meanwhile, the activation of IKK/IκB/NF-κB signal transduction was prevented and the secretion of pro-inflammatory factors was reduced. Ultimately, Aβ deposition and p-tau levels were decreased in APP/PS1 double-transgenic AD mice, and memory and cognitive impairment were improved [60].

An interesting study revealed that PD patient-derived cells had alterations in cellular responses of Nrf2 signaling to sulforaphane [61]. Thus, clinical trials targeting the Nrf2 pathway are warranted to validate the neuroprotective effects of phytochemicals. As shown in Table 1, sulforaphane and broccoli sprout extract have completed clinical trials in autism spectrum disorder and schizophrenia, and dimethyl fumarate has also completed clinical trials in multiple sclerosis. It should be noted that trials in which the water extract of centella asiatica water extract product improved cognitive impairment have been terminated.

## 4. Phytochemicals Target Nrf2 for Cardiovascular Diseases Intervention

Cardiovascular diseases (CVD) are a class of diseases that involve the heart or blood vessels (arteries and veins) including atherosclerosis, coronary heart disease, cardiomyopathy, ischaemic heart disease, heart failure, hypertensive heart disease, inflammatory heart disease, valvular heart disease and myocardial infarction [62]. Cardiovascular diseases remain the biggest cause of deaths worldwide, according to the Global Burden of Disease (GBD) 2016 Study, noncommunicable diseases (NCDs) accounted for about 40% of total, age-standardized global burden of disease in women, and about 50% of total age-standardized global burden of disease in men; CVDs alone accounted for 20% of total burden in women and 24% of total burden in men [63].

The main cause of the majority of cardiovascular diseases comes from complications of atherosclerosis, and oxidized low-density lipoprotein (Ox-LDL) formation under the stimulation of reactive oxygen species which come from several different sources contributes the pathology of atherosclerosis [64]. This is likely to occur at the sites of endothelial damage which are caused by Ox-LDL itself as well as physical or chemical forces and infection. Endothelial cells, smooth muscle cells (SMCs), and macrophages are the sources of oxidants for the oxidative modification of phospholipids [65]. As shown in Figure 3, Ox-LDL can damage endothelial cells and induce the expression of adhesion molecules such as P-selectin, intracellular/vascular cell adhesion molecule-1 (I/VCAM-1) and proinflammatory cytokines such as monocyte chemoattractant protein-1 (MCP-1) and macrophage colony stimulating factor (mCSF). These processes lead to the tethering, activation, and attachment of monocytes and T-lymphocytes to the endothelial cells. Endothelial cells, leukocytes, and SMCs then secrete growth factors, chemoattractants and other proinflammatory cytokines that act on the migration of monocytes and leukocytes into the subendothelial space [66]. Monocytes ingest lipoproteins and morph into macrophages. Macrophages generate reactive oxygen species (ROS), which convert Ox-LDL into highly oxidized LDL, which is, in turn, taken up by macrophages themselves to form foam cells. Foam cells combine with leukocytes to become the fatty streak, and as the process continues foam cells secrete growth factors that induce SMC migration into the intima. SMC proliferation, coupled with the continuous influx and propagation of monocytes and macrophages, converts fatty streaks to more advanced lesions and ultimately to a fibrous plaque that will protrude into the arterial lumen. Later, calcification can occur and fibrosis continues, yielding a fibrous cap that surrounds a lipid-rich core. This formation may also contain dead or dying SMCs. In acute coronary syndromes (e.g., myocardial infarction), when fibrous plaques rupture, the formation and release of thrombi may ultimately occlude vessels [67]. The pathology that oxidative stress causes cardiovascular injury is further confirmed by animal and human studies [68].

Howard first proposed, based on epidemiological studies, that phytochemicals such as plant sterols, flavonoids, and plant sulfur compounds distributed in vegetable and fruit could prevent coronary heart disease [69]. The following investigations further revealed that phytochemicals in fruits and vegetables, including phytoestrogens in soy, hydroxytyrosol in olives, resveratrol in nuts or red wine, lycopene in tomatoes, organosulfur compounds in garlic or onions, isothiocyanates in cruciferous vegetables, monoterpenes in citrus fruits, and polyphenols in teas or wines, could be independently or jointly responsible for the apparent reduction in CVD risk [70]. However, the mechanism underlying the prevention activity caused by phytochemicals remained unclear. Recent studies in animal model and cultured vascular cells have established that isoflavones increase the activity and expression of eNOS, and upregulate expression of detoxifying and antioxidant enzymes genes [71]. Nrf2 has been suggested to be a valuable therapeutic target for cardiovascular disease [72] and it strongly inhibits the initial stage of atherosclerotic progression in the form of leading to the repression of adhesion molecules such as MCP-1 and VCAM-1 [73]. As shown in Figure 3, hydroxytyrosol from olive could protect against oxidative injury in vascular endothelial cells via induction of Nrf2-mediated HO-1 [74]. Ginkgo biloba extract could exert its anti-atherogenesis and vascular protective effects by inducing vascular HO-1 expression that is regulated by the p38-Nrf2 pathway [75]. A relatively short dietary treatment with broccoli sprouts could strongly protect the heart against oxidative stress and cell death caused by ischemia-reperfusion via activating the Nrf2 pathway in rats [76].

Recent studies mainly focus on ischemia-reperfusion. Some phytochemicals can activate Nrf2 and play a protective role in preventing ischemia-reperfusion injury by inhibiting inflammation, oxidative stress and cell death. As shown in Figure 3, astragaloside IV reduces NLRP3 inflammatory-mediated pyrodeath and inflammatory cytokine release, reducing the volume of cerebral infarction and impaired nerve function in rats [77]. Isoliquiritigenin inhibited the activation of the NF-κB pathway, reduced the expression of pro-inflammatory factors, and decreased myocardial infarction size and improved cardiac function in mice [78]. Various biomarkers of oxidative stress increase after tissue ischemia-induced injury, and some phytochemicals can inhibit this trend. Geraniin activates Nrf2 in vitro and in vivo, increases SOD activity, reduces lactate dehydrogenase (LDH) activity, and the contents of MDA, NO and neuronal nitric oxide synthase (nNOS), reduces infarct volume, and reduces neurodeficit score [79]. Chlorogenic acid restored the expression of Nrf2 signaling pathway, alleviated CI/R-induced brain injury, and enhanced learning and spatial memory. It promoted the expression of BDNF and NGF, which was beneficial to nerve recovery and reduced oxidative stress [80]. Luteolin activates sestrin2, thereby upreguating Nrf2, reducing the release of lactate dehydrogenase (LDH) and MDA and alleviating mitochondrial damage, thus improving heart function and myocardial vitality [81]. A final effect of phytochemicals is to prevent cell death from injury. For example, Rehmannioside A improves cognitive impairment and neurological deficits resulting from cerebral artery occlusion and reperfusion by inhibiting iron death [82]. Tanshinone I increases Nrf2 signaling pathway expression, alleviates doxorubicin-induced mitochondrial structural damage, reduces cardiomyocyte apoptosis and improves cardiac function in mice [83]. In human clinical trials, curcumin administration reduced the serum levels of cholesterol and lipid peroxides in 10 healthy human volunteers receiving 500 mg of curcumin daily for seven days [84]. Furthermore, another interventional study with a randomized, double-blind, controlled trial found that the administration of low-dose curcumin also showed a trend of reduction in total cholesterol level and LDL cholesterol level in patients with acute coronary syndrome. Long-term (2–12 months) supplementation with genistein or soy isoflavones showed benefits to the cardiovascular system, including reducing arterial stiffness, lowering blood pressure and improving vascular function [85]. Sulforaphane and broccoli sprout extract have completed clinical trials in autism spectrum disorder and schizophrenia, and dimethyl fumarate has also completed clinical trials in multiple sclerosis. It should be noted that trials in which the water extract of centella asiatica water extract product improved cognitive impairment have been terminated.

## 5. Phytochemicals Target Nrf2 for Cancer Intervention

Cancer is the second leading cause of death, after cardiovascular diseases, in occidental countries. Every year, around 10 million people worldwide are diagnosed with cancer, and approximately 6.2 million die of this disease. Only 5–10% of all cancer cases can be attributed to genetic defects, whereas the remaining 90–95% cancers have their roots in the environmental factors, among which almost 30–35% are linked to diet [86]. It is nearly impossible to prove what caused a cancer in any individual, because most cancers have multiple possible causes, but it is clear that the ratio in a number of cancers is low in the population having diets rich in vegetables, fruits and whole grains while it is high in the population having diets rich in processed or red meats [87].

Cancer development in humans is a multistep, long-term process, in which cancer cells acquired several biological capabilities such as sustaining proliferative signaling, evading growth suppressors, resisting cell death, enabling replicative immortality, inducing angiogenesis, activating invasion and metastasis, reprogramming of energy metabolism, evading immune destruction, genome instability and mutation and tumor-promoting inflammation [88]. Due to cancer pathogenesis being traceable back to that impact cell growth and metastasis, oxidative stress is one of the main causes for carcinogenesis.

Oxidative stress promotes damage to the cell structure, including proteins, lipids, membranes and DNA, thus plays a key role in the development of cancer [89]. As shown in Figure 4, mitochondrial electron-transport chain, proinflammatory cytokines and other oxidizing agents are the prime pathways that generate excess ROS in vivo, leading to several types of DNA damage, including depurination and depyrimidination, single and double-stranded DNA breaks, base and sugar modifications and DNA-protein crosslinks [90]. Permanent modification of genetic material resulting from the oxidative damage is one of the vital steps involved in mutagenesis that leads to carcinogenesis. Stimulation of DNA damage can cause transcription disorder, replication errors and genomic instability. All of these happenings are associated with carcinogenesis [91]. Excess oxidative stress has been considered to promote cancer [92]. Endogenous oncogenic alleles of Ras, Braf and Myc in primary mouse cells increase Nrf2 transcription and decrease intracellular ROS levels [93]; when Nrf2 is over-activated, it can protect cancer cells from oxidative stress and other adverse conditions and reduce the effect of anticancer drugs. Therefore, Nrf2 inhibitors have gradually been found to assist in inhibiting cancer cell proliferation [94]. The first study that found phytochemicals may play positive role in cancer prevention was carried out half a century ago, in which researchers administrated small quantities of phytochemicals decreased the incidence of cancer in rats [95]. From then on, many phytochemicals have been proved gradually to possess the chemoprevention activity by mediating Nrf2-ARE pathway. These phytochemicals can be classified into “blocking agents” who impede the initiation stage and “suppressing agents” which arrest or reverse the promotion and progression of cancer [96]. The well investigated phytochemicals with obvious cancer prevention ability in animal and cell test include sulforaphane from broccoli and wasabi, curcumin from turmeric root, EGCG from green tea, resveratrol from grape, and caffeic acid phenethyl ester from coffee. Additionally, quercetin, myricetin, garlic oranosulfur compounds, lycopene, purpurogallin, avicins widely distributed in fruits and vegetables are also found to have cancer chemoprevention activity [97]. These phytochemicals exert their chemopreventive activity through the induction of Nrf2-dependent adaptive responses including phase II detoxifying enzymes, antioxidant proteins and transporters that protect cells from carcinogens or other stimulations [98]. A549 and HepG2 cells can up-regulate the level of glutathione and resist the antitumor effect of osteomycin, and the Nrf2 in A549 is higher, and the resistance to osteomycin is stronger. Knockdown of Nrf2 and inhibition of Nrf2 expression with luteolin significantly reduced resistance to anticancer drugs [99]. Another study demonstrated that the protein expression of Nrf2 and HO-1 plays an important role in cancer stem cell formation, and the combination of luteolin and the chemotherapy drug paclitaxel enhanced cytotoxicity in breast cancer cells by downregulating Nrf2 expression [100]. Administration of glucosinolate-rich broccoli sprout reduced aflatoxin-caused DNA adducts through induction of GST activity in human clinical trials [101]. Sulforaphane blocked benzo[α]pyrene-evoked forestomach tumors in ICR mice via Nrf2 [102]. Curcumin works as chemosensitizer and radiosensitizer for tumors therapy through activation of Nrf2-mdiated expression of antioxidant enzymes in preclinical trials [103]. Another recent study using microarray to analyze gene expression and regulation pathways in tissue of men with prostate cancer in a randomized clinical trial found that lycopene supplementation could activate Nrf2 pathway [104].

It is noticed that Nrf2 plays dual roles in normal cells and cancer cells. In normal cells, Nrf2-mediated induction of phase II antioxidant proteins and detoxifying enzymes is beneficial to cancer chemoprevention, while in cancer cells, Nrf2-mediated induction of phase II detoxifying enzymes have reported to increase drug resistance and promote proliferation of cancer cells [105]. However, in recent years, it has been proposed that Nrf2 may be a transcription factor with multiple roles, both in tumor suppression and tumor promotion [106]. As shown in Figure 4, astragaloside IV also activates the phosphorylation of Nrf2 and upregulates the expression of HO-1 and NQO1 while inhibiting primary liver cancer in mice [107]. Sulforaphane can inhibit a variety of cancers, which is reflected in various stages of cancer occurrence and development. The key point is to regulate cell homeostasis through the activation of Nrf2, protect cells from DNA damage and promote apoptosis, and regulate cell cycle by anti-metastasis [108]. That is why limited studies were performed on cancer chemoprevention of phytochemical by targeting Keap1-Nrf2-ARE pathway. Thus, it needs to clarify the effects and mechanisms of dietary phytochemicals in cancer cells, although the role of dietary phytochemicals in cancer prevention is clear. Nrf2 has only been investigated in two cancers in clinical trials. Broccoli sprout extract has been used in cigarette smoking-related carcinoma and head and neck cancer studies have been completed. It can inhibit cell carcinogenesis from cigarette smoking, and a new round of clinical trials is now being recruited.

## 6. Phytochemicals Target Nrf2 for Inflammatory Diseases Intervention 

Inflammation is a complex set of interactions among soluble factors and cells, and can arise in any tissue in response to traumatic, infectious, post-ischaemic, toxic or autoimmune injury [109]. Thus, inflammation is at the root of several degenerative disorders, such as autoimmune diseases, rheumatoid arthritis, asthma, emphysema, gastritis, colitis, osteoarthritis, chronic obstructive pulmonary disease, ageing, atherosclerosis, cancer [110]. As shown in Figure 5, inflammation occurs as part of the immune reaction and produces ROS to inactivate the foreign molecules and fight against the invading pathogens. A cascade of cytokine- and chemokine-mediated inflammatory reactions initiate and maintain a host response, involving activation and attraction of immune and non-immune cells. Leukocytes (neutrophils, monocytes and eosinophils), macrophages, lymphocytes, and plasma cells in venous system infiltrate into the disrupted and damaged tissue to recover from infection and to heal [109]. However, cytokines and chemokines persisting at inflammatory sites cause chronic oxidative stress that can mediate subsequent tissue injuries. Oxidative stress activates the redox-sensitive transcription factors such as NF-κB and AP-1, resulting in the production of pro-inflammatory cytokines and chemokines. Excess oxidative stress is also involved in the pathophysiologies of many inflammation-associated disorders. Therefore, the induction of antioxidant proteins and detoxifying enzymes through activation of Nrf2-ARE pathway is essential for the body’s protection against inflammatory tissue injuries [111].

Recent studies confirmed that phytochemicals and their main bioactive compounds could prevent chronic inflammatory related disorders by activation of Nrf2-ARE pathway. As shown in Figure 5, traditional Chinese medicine Fuzi from lateral roots of *Aconitum carmichaeli Debx* could play a beneficial effect on rheumatoid arthritis patients by regulating the Nrf2 pathway [112]. Strawberry extract C effectively neutralized LPS-induced oxidative stress and reduced ROS and NO production by activating Nrf2 in RAW 264.7 macrophages and inhibiting NF-κB signaling [113]. For the bioactive compounds, curcumin mediates its anti-inflammatory effects through induction of phase II enzymes that was confirmed by pharmacodynamics and pharmacokinetics in pulmonary disease in both animals and humans [114]. Sulforaphane in broccoli sprouts could enhance protection of gastric mucosa against oxidative stress in vitro and exerted anti-inflammatory effects on gastric mucosa during H. pylori infection in mice and human through activation of Nrf2 pathway [115].

The potential molecular mechanisms of the anti-inflammatory effect of phytochemical by targeting Nrf2 pathway mainly include regulation of NF-κB, NLRP3 inflammasome, STAT1/IL-6, JAK2/STAT3, etc. As shown in Figure 5, sinomenine in *Sinomenium acutum* [116], Salvianolic acid C [117], are all reported to inhibit NF-κB pathway by targeting Nrf2 to exert their anti-inflammatory effect in multiple cell or animal models. EGCG in green tea has been proved to prevent lupus nephritis in mice by enhancing the Nrf2 signaling pathway and decreasing renal NLRP3 inflammasome activation [118]. Resveratrol attenuates rotenone-induced BV-2 cell activation and M1 polarization by promoting upregulation of STAT1, Nrf2 and SLC7A11, and reduces the production of IL-6, IL-1β and TNF-α pro-inflammatory factors [119]. Salidroside inhibits hypoxic-induced serum and liver pro-inflammatory cytokine release through JAK2/STAT3 mediated pathway, while enhancing the Nrf2-mediated antioxidant pathway activation [120].

Regardless, many other phytochemicals are also reported to play important roles in inflammatory diseases by targeting Nrf2 pathway. Baicalin activates the antioxidant pathway and inhibits the mRNA of the pro-inflammatory cytokines IL-1β, IL-6, MCP-1 and TNF-α, further improving diabetic nephropathy [121]. Many experiments have found that the activation of Nrf2 is a necessary condition for inhibiting inflammation. For example, Nardochino C (a new compound isolated from *Leptococcus officinalis*), baicalin, phenyl isothiocyanate, curcumin, brassicin and red ginseng-derived saponin can inhibit the expression of pro-inflammatory cytokines such as IL-1β, IL-6, MCP-1 and TNF-α in macrophage, but when Nrf2 is knocked out or knocked down, the inhibition effect of these phytochemicals disappeared [121,122,123,124].

The potential use of these phytochemical antioxidants in the treatment of chronic obstructive pulmonary disease (COPD) has been supported by in vitro data, animal models or human preclinical studies [125,126]. During the last two decades, many clinical trials by targeting Nrf2 to treat inflammatory diseases are underway, including obesity, chronic kidney disease, gout and other inflammatory diseases, as shown in Table 1. Among which, broccoli or broccolisprout and its main bioactive compound sulforaphane are the most widely used phytochemicals to act as an Nrf2 inducer, and chronic kidney diseases, obesity and arthritis are the most studied diseases. Most of the clinical trials have been completed and its clinical application is on the way.

## 7. Phytochemicals Target Nrf2-ARE Pathway for Other Chronic Diseases Intervention (Obesity, Ageing and Longevity)

Phytochemicals showed their potency in prevention on many other chronic diseases through activation of Nrf2-ARE pathway. Oxidative stress in accumulated fat appears as an earlier instigator of obesity-associated metabolic syndrome; thus, it is an important target for therapies.

Obesity is a medical condition in which excess body fat has accumulated to an extent and is reaching worldwide unprecedented prevalence in persons of all ages [127]. Obesity causes a series of health problems like metabolic syndrome [128] and likelihood of various diseases, especially including heart disease, type 2 diabetes, obstructive sleep apnea, certain types of cancer, and osteoarthritis [129]. Oxidative stress shows a strong relation to obesity. In brief, as shown in Figure 6, hyperglycemia and excess oxidative stress stimulate adipocyte generation. Mature adipocyte could induce more oxidative stress and secret a series of proinflammatory cytokines such as IL6, PAI-1, MCP-1, and TNF-α, companying inhibition of adiponectin and leptin production to damage tissues and cause chronic inflammation. All of these events finally lead to metabolic syndromes, including diabetes, insulin resistance, atherosclerosis and hypertension. Besides, obesity per se may induce systemic oxidative stress in accumulated fat that, in turn, causes dysregulation of adipocytokines and further develops metabolic syndrome [127,130].

The obesogenic environment of highly palatable foods with hidden fats and sugars can promote metabolic syndrome and obesity, whereas fruit and vegetables with phytochemicals can counteract metabolic syndrome and obesity [131]. For example, isoflavones and lignans have been proved to exert anti-obesity activity in nutritional intervention studies in both animals and humans [132]. Genistein, conjugated linoleic acid (CLA), docosahexaenoic acid, epigallocatechin gallate, quercetin, resveratrol and ajoene affect adipocytes during various stages of the adipocyte life cycle, resulting in either inhibition of adipogenesis or induction of adipocyte apoptosis which decreases lipid accumulation and induces lipolysis [133]. Moreover, polyphenol-rich grapes can target obesity-induced oxidative stress through activation of Nrf2-ARE pathway [134].

The role of Nrf2 in longevity has received much attention. For example, Nrf2 is identified as a mediator of caloric restriction [135], and as an effector of longevity signals, providing new therapeutic perspectives [136,137]. However, several blockbuster animal studies have shown that Keap1 is the key factor affecting lifespan. One study found that p62(-/-) mice had accelerated aging and impaired mitochondrial function, and further studies found that the role of p62 in this process was to promote the autophagic degradation of Keap1 and indirectly promote the expression of Nrf2 [138]. In addition, studies on nude mice and nine other rodent species found that maximum life potential was not related to the protein level of Nrf2 itself, but was negatively correlated with Keap1 [139]. The molecule mechanism may be that Nrf2, when functioning normally, promotes NQO1 expression and limits peroxide production to maintain mitochondrial integrity and energy metabolism homeostasis to ensure that the life span is not shortened [140]. In addition, transcription factors such as SKN-1/Nrf2 and DAF-16/FOXO can up-regulate lysosomal gene expression, which plays important roles in prolonging life span by maintaining the normal function of lysosomes [141]. Another study showed that Nrf2, although not directly involved in longevity signaling, is essential for the maintenance of metabolic and protein homeostasis, and Nrf2 knockdown leads to a shortened lifespan in mice [142]. This suggests an indirect role for Nrf2. While Nrf2 activation is strongly associated with longevity extension, it also cannot be overexpressed. High Nrf2 expression levels also lead to death or other pathologies, but inhibition of insulin/IGF-like signaling and diet suppression mildly activate Nrf2 and prolong the lifespan of flies [143].

Phytochemicals also showed their potential effects on calorie restriction and longevity accompanied by Nrf2 activation. Caenorhabditis elegans is the most commonly used model organism in longevity studies. As shown in Figure 6, Ginsenoside extract, 6-hydroxyflavone, baicalein, epigallocatechin gallate (EGCG) and epigallocatechin gallate (ECG), oleuropein activate antioxidant regulating and longevity signaling pathways, For example, Nrf2/SKN-1, SIRT1/SIR2.1 and FOXO/DAF-16 signaling pathways prolong the lifespan of C. elegans [144,145,146,147].

In addition, damage to cells by external factors such as the environment can also lead to senescence and reduce life span, such as ultraviolet radiation. Galangin can activate SIRT1/PGC-1α/Nrf2 signaling pathway [148], and cordycepin can bind and activate AMPK/Keap1/Nrf2 to increase the expression of antioxidant enzymes, thereby preventing cell senescence [138].

It is clear that, since the discovery of Nrf2-ARE pathway, enormous advances have been made over the past decade in understanding the mechanisms of action of phytochemicals. Obviously, phytochemicals can play a central role in the prevention of a series of oxidative stress-related chronic diseases by regulating the Nrf2-ARE pathway. However, most of the data has been undertaken in cells or mice, not only because of the nearsightedness of company researchers who only paid main attention to limited biomarkers, but also because of the neglect and deficiency of study of crosstalks between the Nrf2-ARE pathway and other disease related signaling transduction pathways. Phytochemicals definitely contribute to human health, so there is now an urgent need to translate cell or animal-based knowledge into humans. There were only two clinical trials of aging and Nrf2, one of which was that sulforaphane enhanced Nrf2 signaling in elderly people, but the other trial of UV irradiation causing aging was terminated for administrative reasons.

## 8. Conclusions and Future Perspectives

During the last decades, the comprehensive molecular mechanism of Keap1-Nrf2-ARE pathway activation caused by phytochemicals is emerging slowly, which supply the molecular basis for future application. Crosstalk between Keap1-Nrf2-ARE pathway and other signaling pathways endows it high complexity and significance in the multi-function of phytochemicals. Accumulating data in cell and animal models revealed that phytochemicals target the Keap1-Nrf2-ARE antioxidant pathway to exert their potent and promising ability in the prevention of a series of chronic diseases. However, limited human data makes an urgent need to open the new field of phytochemical-original supplement application in human chronic disease prevention, and comprehensive evaluation is seriously needed for the dual role of Keap-Nrf2-ARE pathway activation, especially the sustained activation of Nrf2 appears to favor the progression of cancer and other diseases. Obviously, phytochemicals targeting the Keap1-Nrf2-ARE pathway for chemoprevention will lead to the rapid development of phytochemical drugs with low toxicity and high efficiency for chronic disease intervention.

## Figures and Tables

**Figure 1 antioxidants-12-00236-f001:**
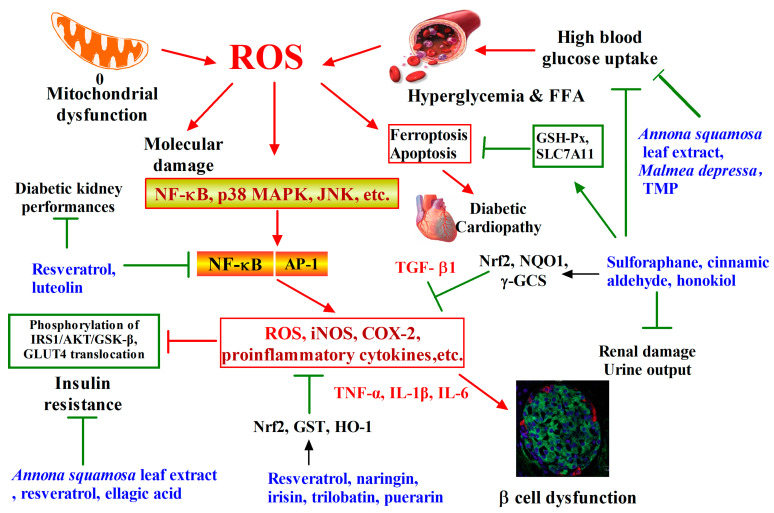
Phytochemicals target Nrf2 for diabetes intervention. Mitochondrial dysfunction and hyperglycemia lead to the massive production of ROS, which triggers molecular damage, inflammation, ferroptosis, insulin resistance, and β-cell dysfunction. A variety of phytochemicals activate Nrf2 signaling pathway, inhibit ROS levels, ferroptosis and inflammation, restore insulin signaling pathway transduction and protect β cells from injury.

**Figure 2 antioxidants-12-00236-f002:**
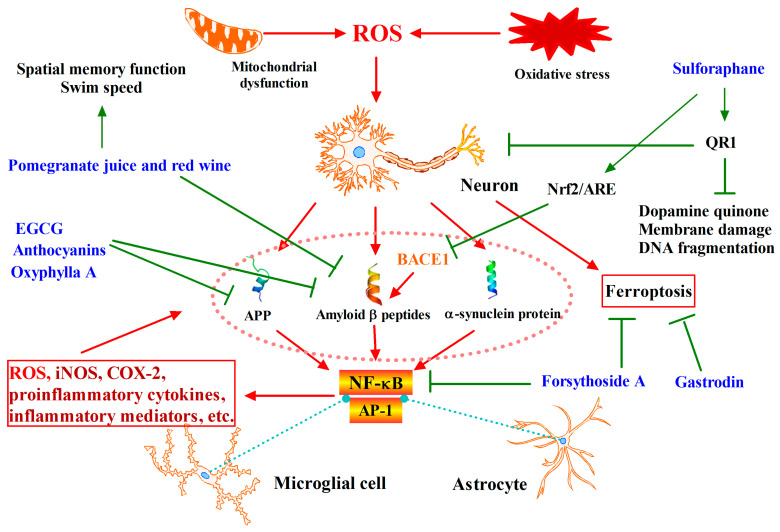
Phytochemicals target Nrf2 for neurodegenerative disease intervention. ROS and inflammatory factors promote the accumulation of harmful proteins such as APP, Amyloid β peptides, α-synuclein protein, and induce ferroptosis of cells, leading to nerve cell damage. A series of phytochemicals are reported to prevent neuron cell damage by blocking harmful protein production and ferroptosis, thereby preventing neurodegenerative diseases.

**Figure 3 antioxidants-12-00236-f003:**
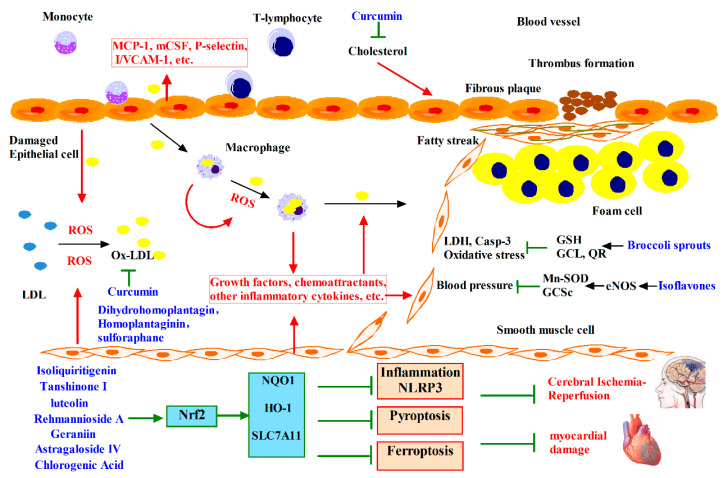
Phytochemicals target Nrf2 for cardiovascular diseases intervention. Ox-LDL can damage endothelial cells and induce the expression of proinflammatory cytokines. Monocytes take up lipoproteins and metamorphoze into macrophages, which can produce reactive ROS, and subsequently convert Ox-LDL to highly oxidized LDL and form foam cells, combine with leukocytes to form fatty streaks and eventually form fibrous plaques that protrude into the arterial lumen, leading to vascular obstruction. A variety of phytochemicals are found to activate the Nrf2 signaling pathway to ameliorate oxidative stress and prevent fibrous plaque formation. Phytochemicals attenuate heart and brain damage after ischemia-reperfusion injury by inhibiting inflammation, apoptosis and ferroptosis.

**Figure 4 antioxidants-12-00236-f004:**
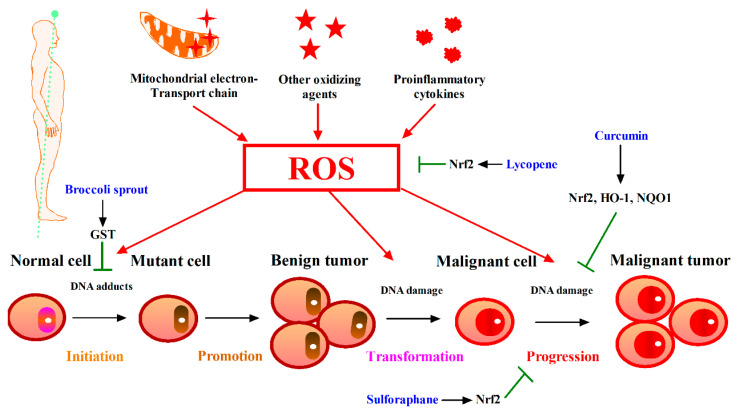
Phytochemicals target Nrf2 for cancer intervention. ROS can promote the transformation of normal cells into cancer cells. Many phytochemicals are reported to prevent cell transformation by ameliorating oxidative stress environment and preventing DNA damage in the early stage of carcinogenesis.

**Figure 5 antioxidants-12-00236-f005:**
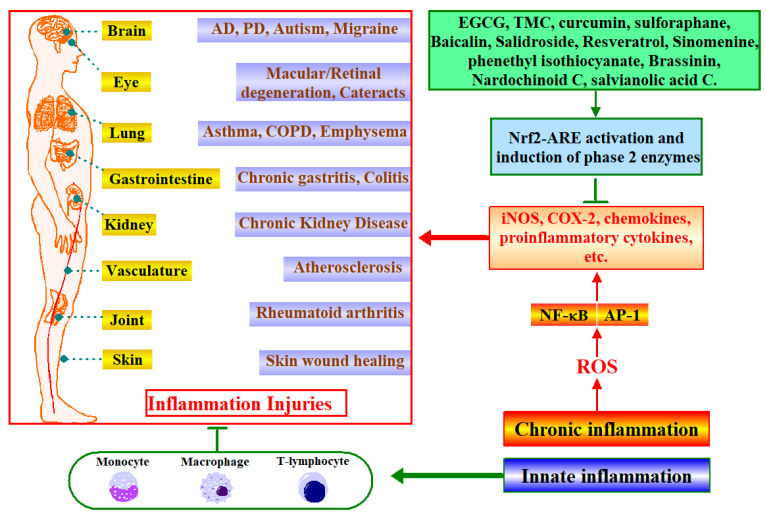
Phytochemicals target Nrf2 for inflammatory diseases intervention. Chronic inflammation produces ROS, which leads to the release of more inflammatory cytokines and inflammatory damage in various tissues and organs. A large number of studies have shown that phytochemicals can activate Nrf2-ARE signaling pathway and induce phase II enzymes, thereby inhibiting the expression of pro-inflammatory factors and preventing inflammatory damage.

**Figure 6 antioxidants-12-00236-f006:**
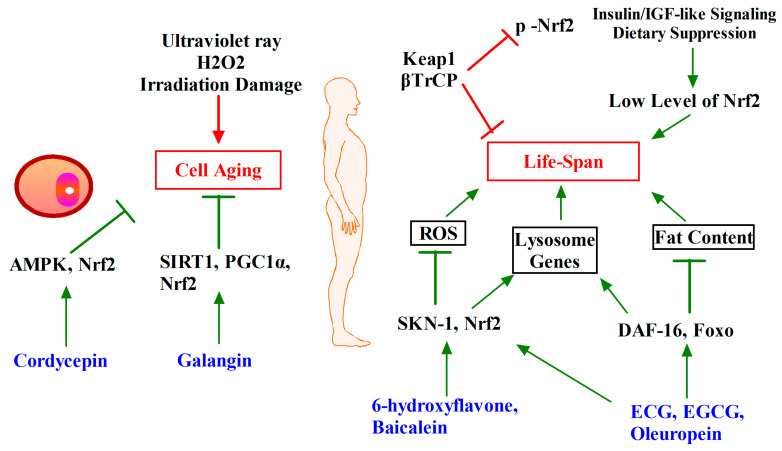
Phytochemicals target Nrf2 for ageing and longevity intervention. Phytochemicals inhibit cell senescence caused by external factors by activating Nrf2. In the in vivo setting, inhibition of Keap1 expression, with appropriate activation of Nrf2, can prolong lifespan. Phytochemicals can activate Nrf2 and other life-related signaling pathways, reduce ROS production, up-regulate lysosomal gene expression, reduce fat content, and ultimately extend the life span of animals.

**Table 1 antioxidants-12-00236-t001:** Summary of the Nrf2 inducer/activators drug discovery in chronic diseases registered in ClinicalTrials.gov of NIH (https://www.clinicaltrials.gov/, accessed on 18 December 2022).

Disease	Natural Compounds	Development	Content	ClinicalTrials.gov Identifier
**Diabetes**				
Pre-diabetic	*Tropaeolum majus* L.	Completed	Tropaeolum majus l intake and biochemical parameters in pre-diabetic subjects in bogota colombia	NCT05346978
Diabetes mellitus, non-insulin-dependent	Sulforaphane	Completed	Clinical trial with broccoli sprout extract to patients with type 2 diabetes	NCT02801448
Insulin sensitivity, overweight	Chickpea and black bean	Recruiting	Pulses consumption and its role in managing systemic inflammation, insulin sensitivity and gut microbiome in human	NCT04267705
**Neurodegenerative diseases**				
Major depressive disorder	Sulforaphane	Recruiting	A comparative study on efficacy and safety of add-on sulforaphane or rTMS to escitalopram for major depressive disorder with poor response to initial treatment	NCT05145270
Alzheimer disease	Hydralazine hydrochloride	Recruiting	Effect of hydralazine on alzheimer’s disease	NCT04842552
Cognitive impairment	Centella asiatica water extract product	Terminated	Pharmacokinetics centella asiatica product in mild cognitive impairment	NCT03937908
Autism spectrum disorder	Sulforaphane	Completed	Sulforaphane treatment of children with autism spectrum disorder (ASD)	NCT02561481
Schizophrenia	Broccoli sprout extract	Completed	An open study of sulforaphane-rich broccoli sprout extract in patients with schizophrenia	NCT01716858
Multiple sclerosis	Dimethyl fumarate	Completed	Pharmacokinetics of DMF and the effects of DMF on exploratory biomarkers	NCT02683863
Friedreich ataxia	Resveratrol	Active, not recruiting	Micronised resveratrol as a treatment for friedreich ataxia	NCT03933163
**Cardiovascular diseases**				
Diastolic dysfunction	Sulforaphane	Recruiting	Prevention of age-associated cardiac and vascular dysfunction using avmacol ES	NCT05408559
Coronary artery disease	Curcumin	Not yet recruiting	Effects of curcumin on markers of cardiovascular risk in patients with CAD	NCT04458116
Anthracycline related cardiotoxicity	Sulforaphane	Recruiting	Protective effects of the nutritional supplement sulforaphane on doxorubicin-associated cardiac dysfunction	NCT03934905
**Cancer**				
Head and neck cancer	Broccoli Sprout Extract	Completed	Broccoli sprout extract in preventing recurrence in patients with tobacco-related head and neck squamous cell cancer	NCT03182959
Cigarette smoking-related carcinoma	Broccoli sprout/broccoli seed	Completed	Broccoli sprout/broccoli seed extract supplement in decreasing toxicity in heavy smokers	NCT03402230
Cigarette smoking-related carcinoma	Broccoli sprout/broccoli seed extract	Not yet recruiting	Testing the effect of the broccoli seed and sprout extract on the cancer causing substances of tobacco in heavy smokers	NCT05121051
**Inflammatory diseases**				
Cystic fibrosis	Broccoli sprouts	Completed	Effect of sulforaphane in broccoli sprouts on Nrf2 activation	NCT01315665
Osteoarthritis	rosemary, ashwagandha, and Sophora japonica	Terminated	PB125, osteoarthritis, pain, mobility, and energetics	NCT04638387
Rheumatoid arthritis		Completed	A new mode of action of anti-TNF, reverse signaling, in rheumatoid arthritis	NCT03216928
Overweight, obesity	Strawberry and red raspberry	Recruiting	Berries, inflammation, and gut microbiome	NCT04100200
Obesity, inflammation	Grape powder	Completed	The effects grapes on health indices	NCT01674231
Inflammation	Sulforaphane	Completed	The protective effect of sulforaphane on chronic low-grade inflammation in healthy participants	NCT05146804
Overweight, obesity	Mango	Recruiting	Anti-inflammatory effect and associated mechanisms of mango consumption	NCT04726293
Gout	Tart cherry extract	Suspended	Pharmacokinetics and pharmacodynamics of anthocyanins	NCT03650140
Chronic renal insufficiency	Resveratrol	Completed	Resveratrol’s effects on inflammation and oxidative stress in chronic kidney disease	NCT02433925
Chronic kidney disease	Sulforaphane	Recruiting	Effects of sulforaphane for patients with chronic kidney disease	NCT04608903
Chronic kidney diseases, inflammation	Microcapsules with turmeric and propolis	Active, not recruiting	Effects of microencapsulated propolis and turmeric in patients with chronic kidney disease	NCT05183737
Renal insufficiency	Cranberry	Completed	Effects of cranberry supplementation on chronic kidney disease patients	NCT04377919
Chronic kidney diseases	Tocotrienol rich fraction	Active, not recruiting	Effects of supplementation with tocotrienol on chronic kidney disease patients	NCT04900532
Chronic kidney disease	Dark chocolate	Completed	Chocolate for patients with chronic kidney disease	NCT04600258
Chronic kidney disease	Curcumin	Recruiting	Effects of curcumin supplementation in patients with chronic kidney disease on peritoneal dialysis	NCT04413266
Chronic kidney disease	Curcumin	Active, not recruiting	Effects of curcumin in patients in chronic kidney disease	NCT03475017
Sickle red blood cell, fetal hemoglobin, oxidative stress	Broccoli sprouts	Completed	Effect of broccoli sprouts homogenate on SRBC	NCT01715480
**Aging**				
Aging problems	Sulforaphane	Recruiting	Treatment strategy to enhance Nrf2 signaling in older adults	NCT04848792
Aging	Sulforaphane	Terminated	Effect of topical sulforaphane on skin fragility seen in skin aging and with ultraviolet exposure	NCT03126539

## Data Availability

The data used to support the findings of this study are available from the corresponding author upon request.

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
