# Peer review of "The Regulatory Effect of Phytochemicals on Chronic Diseases by Targeting Nrf2-ARE Signaling Pathway"

_antioxidants, 2023, doi:10.3390/antiox12020236_

Round 1

Reviewer 1 Report

The review focused on the possible mechanism of Nrf2-ARE pathway activation by natural phytochemicals in the prevention of chronic diseases and the regulation of oxidative stress. Some place still need further improving.

1Line 2, “Nrf2-ARE pathway” in the title, is it accurate

2Line 32-75In Introduction, the logic of the second and third paragraphs is not clear, and the last paragraph should include the purpose of this review.

3The authors should add more detailed text description and summary of the contents of figures and tables.

4、 Line 272, “Table 2” should be “Table 1”.

5The expression of “nrf2” and “Nrf2” should be unified.

6Many Phytochemicals have impact on the chronic releases without regulating Nrf2 pathway in this review. Therefore, the Nrf2 pathway mainly exists at the speculative level, which makes its title not convincing enough.

Author Response

1. Line 2, “Nrf2-ARE pathway” in the title, is it accurate?

Response: Thank you for your reminder and the “Nrf2-ARE pathway” in the title is revised as “Nrf2-ARE signaling pathway”.

2. Line 32-75,In Introduction, the logic of the second and third paragraphs is not clear, and the last paragraph should include the purpose of this review.

Response: It is a good suggestion and we have made corresponding modifications as shown in the last three lines of Introduction section accordingly and seriously, many thanks! According to the logic problem, we have revised it accordingly, that is the second paragraph aims to introduce the activation of Nrf2 pathway is critical in ROS extinguish and chronic disease prevention, and the third paragraph is designed to point out the antioxidant and anti-inflammatory effects of phytochemicals is attributed to Nrf2 activation.

3. The authors should add more detailed text description and summary of the contents of figures and tables.

Response: Thanks a lot for your suggestions, and we have revised the text accordingly, as shown in the manuscript.

4. Line 272, “Table 2”should be “Table 1”.

Response: Thanks a lot for your gentle reminder. As suggested, we have deleted “Table 2”.

5. The expression of “nrf2” and “Nrf2” should be unified.

Response: Thank you for the reminder, and we have unified the expression of“Nrf2”.

6. Many phytochemicals have impact on the chronic releases without regulating Nrf2 pathway in this review. Therefore, the Nrf2 pathway mainly exists at the speculative level, which makes its title not convincing enough.

Response: We have carefully examined the manuscript and deleted the several references which have not mentioned Nrf2 activation according to the criterion of the present title, as shown in the manuscript.

Reviewer 2 Report

1.      I have to admit that the subject is quite an interesting review with enough data generated.

2.      In my interpretation, this paper is good and well-written.

3.      Authors must follow the Journal guideline to cite the reference, [in numerical], suggested ignoring old references.

4.      Although, the manuscript would greatly improve through proofreading for a few syntax errors.

5.      Abbreviations must be defined at first mention and used consistently thereafter in the entire manuscript. Revise the entire manuscript. Eg. LN 80:  ROS

6.      Figures were looks good, is derived from other sources? Quite confusing not clear. Mentioned reference without year, similarly in all Figures? He and Qin et al?

7.      Plant names, species should be in italics, check and revise in entire manuscript. Eg. LN 107

8.      Table 1 should cite the proper references.

9.      Table 2 was missing.

10.  LN 505-507 [cite appropriate references]

11.  Encourage to submission of a graphical abstract.

Author Response

  1. I have to admit that the subject is quite an interesting review with enough data generated.

Response: Thank you for your comments!

  1. In my interpretation, this paper is good and well-written.

Response: Thank you for your comments!

  1. Authors must follow the Journal guideline to cite the reference, [in numerical], suggested ignoring old references.

Response: Thank you for your gentle reminder, and we have carefully revised them in accordance with the journal reference format.

  1. Although, the manuscript would greatly improve through proofreading for a few syntax errors.

Response: Thanks a lot for your suggestion and we have double check the grammar and spell errors and made necessary revisions already and seriously.

  1. Abbreviations must be defined at first mention and used consistently thereafter in the entire manuscript. Revise the entire manuscript. Eg. LN 80:  ROS

Response: Thank you for the gentle reminder and we have revised the abbreviations accordingly and carefully.

  1. Figures were looks good, is derived from other sources? Quite confusing not clear. Mentioned reference without year, similarly in all Figures? He and Qin et al?

Response: “He and Qin et al” means the first author and the correspondence author of the present manuscript, which may help the reviewers or authors easy to read and understand. We can delete it if it causes any confusion. Thank you for your reminder!

  1. Plant names, species should be in italics, check and revise in entire manuscript. Eg. LN 107

Response: Many thanks! We have revised it accordingly.

  1. Table 1 should cite the proper references.

Response: Table 1 is the summary of the Nrf2 inducer/activator drug discovery in clinical trials registered in NIH website, which is attached in the manuscript, and the references have revised as standard description as NCTXXXXXXXX.

  1. Table 2 was missing.

Response: We deleted Table 2 in the present review, sorry for the mistake!

  1. LN 505-507 [cite appropriate references]

Response: Revision has been made, many thanks!

  1. Encourage to submission of a graphical abstract.

Response: Thanks a lot for your good suggestion, we plan to try but time is limited.
